# Lab-on-a-Chip Technologies for Microgravity Simulation and Space Applications

**DOI:** 10.3390/mi14010116

**Published:** 2022-12-31

**Authors:** Aditya Vashi, Kamalalayam Rajan Sreejith, Nam-Trung Nguyen

**Affiliations:** Queensland Micro- and Nanotechnology Centre, Griffith University, Nathan, QLD 4111, Australia

**Keywords:** microgravity simulation, Lab-on-a-Chip (LOC), space application, clinostats, rotating wall vessel (RWV), random position machine (RPM), diamagnetic levitation, CubeSat, acoustic levitation, levitation

## Abstract

Gravity plays an important role in the development of life on earth. The effect of gravity on living organisms can be investigated by controlling the magnitude of gravity. Most reduced gravity experiments are conducted on the Lower Earth Orbit (LEO) in the International Space Station (ISS). However, running experiments in ISS face challenges such as high cost, extreme condition, lack of direct accessibility, and long waiting period. Therefore, researchers have developed various ground-based devices and methods to perform reduced gravity experiments. However, the advantage of space conditions for developing new drugs, vaccines, and chemical applications requires more attention and new research. Advancements in conventional methods and the development of new methods are necessary to fulfil these demands. The advantages of Lab-on-a-Chip (LOC) devices make them an attractive option for simulating microgravity. This paper briefly reviews the advancement of LOC technologies for simulating microgravity in an earth-based laboratory.

## 1. Introduction

Recent advancements in micro and nanotechnology led to their successful implementation in biomedical, biochemical, pharmaceuticals, chemical, and biotechnological applications. Terry et al. demonstrated the first miniaturized gas chromatograph in 1979 [1]. In the subsequent decade, researchers focused on developing miniaturized components such as valves, pumps, and sensors for their integration into a single system. At the end of the 1990s, Manz et al. proposed the concept of “Miniature Total Chemical Analysis Systems”, integrating most chemical analysis protocols into a single chip [2]. Later, this concept was applied to the areas mentioned above and became widely popular as the “Micro Total Analysis System (μTAS)” or “Lab-on-a-chip (LOC)”, which incorporates laboratory processes and functions into a single chip that further extends to biological applications [3].

Due to miniaturization, LOC devices offer multiple advantages over conventional laboratory-based systems. The key advantage is portability, enabled by the reduced size of the devices [4]. Moreover, the small size minimizes the amount of sample and reagent needed. Miniaturization also increases the surface-to-volume ratio and controls the reaction efficiently in terms of outcome and time [5]. In addition, well-developed fabrication technologies and the small size of LOC devices reduce the overall cost of running experiments [6]. However, LOC systems still have their limitations in some respects. The fabrication of LOC devices requires an expensive cleanroom facility and a skilled workforce [7]. Moreover, manipulating and controlling continuous flow in the devices require complex channel geometries, pumps, tubing, and valves [8]. In addition, clogging can occur in the microchannels if the sample contains solid particles [8]. 

Droplet-based digital microfluidics (DMF) offers the same advantages as a continuous flow LOCs system without needing microchannels and fluid handling components. Pollack et al. reported the first rapid manipulation of individual droplets [9]. The movement of droplets can be achieved with electrowetting or electrowetting-on-dielectric or thermal approaches [10,11]. To date, DMF has found applications in chemistry, biochemistry, biology, life science, and medicine. While the uptake of DMF technology has been increasing, it is still a relatively new technology and is only available to a few research labs [12]. Moreover, DMF technology still requires the fabrication of a chip, which follows the same procedures as LOC. Lately, researchers are adapting three-dimensional (3D) printing technologies to solve the cost and skilled workforce issues of LOC’s conventional fabrication [13]. Because 3D printing is easy to learn, automated, and provides high resolution and throughput with less fabrication time [14]. 

Recently LOC and DMF technologies have started attracting attention from space research communities. The space environment lacks gravity and contains extreme temperature, ionizing radiation, as well a vacuum [15]. Gravity creates the acceleration of mass on earth and contributes to shear forces, hydrostatic pressure, and sedimentation [15]. Many mass and heat transfer mechanisms, such as free convection, do not work without gravity, relying on concepts such as capillarity, magnetic, or electric field [16]. Zero gravity is only possible in space [17]. Therefore, space experiments are mainly conducted in the international space station (ISS) with minimal gravity. This small gravity is known as microgravity [17]. In ISS, the quality of microgravity is determined by the g-jitters [17]. G-jitters occur due to the vibration of a running machine or onboard human movement. 

Microgravity affects the function and physiology of the human body and microorganisms [18]. Research in a microgravity environment can help to understand the unique behavior of living organisms and open new pathways for developing drugs and vaccines [18]. Moreover, microgravity research can be extended to earth-based chemical operation, which offers advanced chemistry research in a less contaminated environment [19]. Figure 1 shows the important biological, and chemical applications are benefited from microgravity research.

However, running experiments in space face serious challenges. First, advanced equipment is required to address the extreme conditions [20]. Second, the overall running cost of the experiment can be exorbitant [20]. Third, to isolate the microgravity effect from other parameters, these parameters should remain constant, which is difficult to control due to the extreme conditions of the space environment [20]. Last, launching the experiment setup into space, running the operation, collecting data, and verifying the results can take a long time [20]. Apart from these challenges, this opportunity is not available to the broad scientific community, hence restraining the growth of research activities in this field.

Multiple facilities have been developed to recreate the microgravity environment and to address the above problems. Drop towers [21,22], sounding rockets [23,24], and parabolic flights [25,26] are ground-based facilities that can solve time and control issues but require a large budget and significant space on the ground. CubeSat, a satellite mainly built for scientific research, is attractive to many scientists because of its small size and the same environment as ISS [27]. As CubeSats are controlled from the ground, the experiments conducted in space should be accurate and precise. Under this condition, even a minor mistake may result in a failed experiment. It should be noted that none of the facilities mentioned above are laboratory-based. Hence, only a limited number of experiments can be carried out, and the project relies on a facility provider.

Clinostats, rotating wall vessels (RWV), random positioning machines (RPM), and diamagnetic levitation can accommodate microgravity condition in an earth-bound laboratory. Laboratory-based devices are easy to develop, cost-effective, and can be modified to incorporate LOC technologies. Thus, the first section of this paper reviews the laboratory-based devices for microgravity simulation and the implementation of LOC technologies in these devices. Though CubeSats are not laboratory-based, their advantages for microgravity research with the help of LOC devices are attracting interest from the research community. The second section focuses on the CubeSat technology. The third section explores new possibilities to simulate microgravity in the laboratory. Finally, the paper concludes with a summary.

## 2. Lab-on-a-Chip Technologies in Conventional Simulated Microgravity Environment

As mentioned in the introduction, concepts such as clinostats, RWV, RPM, and diamagnetic levitation have been developed to simulate microgravity in the laboratory. This section provides insight into these concepts and their implementations. We first discuss each concept of microgravity simulation. We then present them in detail. Finally, we review recent works utilizing LOC technologies.

### 2.1. Clinostat

Clinostat eliminates the motion of a particle by continuous rotation, which nullifies the effect of gravity [28]. The rotation axis of the clinostat is perpendicular to the gravity vector. Figure 2 illustrates how a high-speed clinostat mimics the condition of microgravity [29]. As shown in Figure 2a, sedimentation of particles occurs under the earth gravity. In microgravity, particles are distributed homogenously with no movement in the liquid (Figure 2b). Rotation of the clinostat starts the circular rotation of the suspended particle. Increasing the speed of clinostat reduces the circular path of the suspended particle (Figure 2c). With a high enough speed, the circular path of the particle becomes negligible and rotates on its own axis, preventing sedimentation. Due to rotation, particles continuously change the gravity vector direction and experience free fall conditions [30]. Clinostat was first used to study plant gravitropism, where the rotation speed was maintained around 1–2 rpm [31]. High-speed rotation between 50–100 rpm is required to investigate the effect of microgravity on mammalian cells and single-cell organisms [31]. 

Clinostat can be divided into five types according to design, configuration, and sample containers. In pipette/cuvette clinostats (Figure 3a), pipettes, tubes, or cuvettes hold the subject [32]. The pipette clinostat developed by the German Aerospace Centre (DLR, Cologne, Germany) can hold up to 10 pipettes and adjust speeds between 0 to 90 rpm [32]. Most experiments with pipette clinostat have been carried out on suspended cell cultures, including mammalian immune cells and stem cells, which are rotated at 60 rpm. The rotation time depends on experiments [32,33,34,35]. Fixations can be done while rotating the pipette, so that sample can be directly transferred to a microscope for analysis [36]. Pipette clinostat cannot be used for adherent cells due to the small diameter of the pipette or cuvette [37]. 

Eiermann et al. adapted slide-flask (Figure 3b) instead of pipettes for adherent cell culture [37]. This technique has been used for investigating the behavior of cancer cells in microgravity environments [38,39,40,41]. In a slide-flask clinostat, slides are kept in the center of the rotation axis for better microgravity simulation. Similar to pipette clinostat, the sample is analyzed after the rotation [42]. A submersed clinostat (Figure 3c) was developed to study the effect of microgravity on underwater organisms [43]. In this clinostat, tubes are submerged in water and rotated underwater to simulate microgravity [44]. Studies showed that aquatic organisms are also affected by the gravity of the earth [45,46,47].

None of the above clinostats can examine the sample during the rotation, limiting the understanding of real-time behavior of the cell culture. Horn et al. developed a portable photomultiplier clinostat (PMT) [48]. In PMT, a photomultiplier tube is connected to a clinostat [48]. The photomultiplier tube amplifies and detects the number of photons emitted by biological samples [48]. The PMT provides real-time observation of suspended cells during the rotation, which is impossible with cuvette clinostat [49]. Besides PMT clinostat, a horizontally positioned microscope combined with a clinostat known as a microscope clinostat can also provide online measurement [50]. The microscope clinostat is attached to a video camera and rotates around its optical axis [50]. However, rotation induces mechanical vibration, causing a disturbance with the microscope. Moreover, these conventional clinostats require more lab space and might not be able to accommodate a LOC system.

Yew et al. developed clinostat time-lapse microscopy (CTM) with compact size, lower cost, and more control [51]. CTM device consist of a stepper motor and a gearbox arrangement for rotation. The device can hold any microfluidic device with a format of a standard microscope slide [51]. This platform is also known as clinochip. Clinochip is halted for 20-the 30s each hour for taking images, allowing for a time-lapse analysis of cell growth [51]. Luna et al. used this device later to observe the effect of angular rotation on stem cells [52]. Since the development of CTM, no other research group has implemented LOC technology in clinostats. As clinostats are affordable and easy to develop compared to other microgravity simulators, there is room for further development of clinostats with integrated LOCs.

### 2.2. Rotating Wall Vessels

A Rotating Wall Vessel (RWV) or rotating cell culture system (RCCS) bioreactor works in the same way as 2D-clinostats [53]. Initially, the primary purpose of developing RWV was to replace conventional bioreactors and to protect the cell culture from high shear stress and turbulence during the launch and landing of the space shuttle [54]. However, the lack of sedimentation of cells and microcarrier during fluid rotation opens up new applications of RWV as a microgravity simulator on earth, particularly for biological processes [55]. 

The first RWV was developed by the National Aeronautics and Space Administration (NASA) [55]. In RWV, a cell culture medium is held in a vessel with no headspace and rotated around a horizontal axis [55]. A silicone membrane is placed at the center of the vessel for oxygenation. Air is circulated through an external pump to prevent air bubbles [55]. In contrast to a 2D-clinostat, RWV has a larger rotating vessel and supply oxygenation system for reaction purposes [56]. Moreover, the circular rotation of particles in clinostats has to be reduced as much as possible, while in RWV, circular rotation is required for agitation between the cell and the microcarrier [56]. 

Following the demonstration of the first RWV (Figure 4a), which is known as the Slow Turning Lateral Vessel (STLV), NASA reported the High Aspect Ratio Vessel (HARVs) (Figure 4b) [57]. The vessel of HARV is shorter and has a wider diameter than STLV [57]. Prewett et al. compared multiple cell growth functions in both RWVs and concluded that better results are achieved with HARV [57]. The main reason is the air exchange membrane at the back of the cylinder, which supplies more oxygen to growing cells [58]. 

Due to the lack of convection in space, mere rotation of the vessel is not enough for the perfusion of gas and nutrients. To solve this problem, NASA configures the design of STLV, which is known as the Rotating Wall Perfused Vessel (RWPV) (Figure 4c). In RWPV, additional rotation is given to the coaxial oxygenator with a vessel, which introduces a little more shear stress and provides better perfusate mixing [59,60]. Like other clinostats, RWVs are used to simulate microgravity for diverse types of cells [61], aquatic organisms [62], and microbes [63,64]. However, the quality of microgravity in RWV is lower compared to clinostat due to particle rotation.

RWV finds the most promising application in tissue engineering because 2-D in vitro cell cultures do not behave as in vivo tissue [65]. Cells settling on the surface restrict their freedom to growing and to reach their optimal 3D form. RWV prevents sedimentation, provides growth conditions with less shear stress and turbulence, simulating microgravity, which is the best environment for forming organoids [66]. Despite these advantages, RWV is yet to be benefited from recent technological advances. 

The primary technology that can be implemented for RWV is 3D printing [67]. Parts of RWV build by 3D printing can significantly reduce the overall cost of the device. Moreover, reducing the number of parts to be assembled can further shrink the size of the final device [67]. Qian et al. developed a custom 3D spinning bioreactor for human brain stem cell organoids [68]. Moreover, Wang et al. reported an organ-on-a-chip approach for the same brain cell, which can be supported for a long time in a simple, low-cost, easy-to-operate chip [69]. The development of this on-chip bioreactor indicates the possible implementation of this approach in RWV for space applications.

### 2.3. Random Positioning Machine

While the clinostat and RWV rotate around a single axis, the random position machine (RPM) (Figure 5a) rotates on two gimbal-mounted frames perpendicular to each other with independent motors [70]. A biological system requires a certain time to sense the gravity vector [70]. If the rotation direction changes continuously, organisms cannot sense the gravity vector and behave as if they are under microgravity conditions [70]. RPM rotates fast enough to counteract the gravity vector and slow enough so that acceleration forces do not become dominant [71]. RPM can be considered as 3D-clinostat if both frames are rotated in the same direction at the same speed. Nevertheless, the direction and speed are kept random, so biological samples do not adapt to a given pattern, continuously reorientate, and over time cause an average gravity vector approaching zero.

Hoson et al. developed the first RPM device for plant research as 2D clinostat could not influence the growth of some plant roots [73,74]. To check the effect of gravity, the RPM is rotated with three operation modes. At a constant speed of both motors (Figure 6a), plant material cannot reverse the motion direction, which cannot compensate for the unilateral influence of gravity [73]. This can be overcome by rotating the motors at a speed ratio of 1:2. In this configuration, the plant material reverses its motion direction, but only moves on a given path shown in Figure 6b [73]. Actual simulated microgravity is achieved when both frames are moved randomly using a random number table (Figure 6c) [73]. Moreover, the speed of the motor also changes randomly between 2 rpm to −2 rpm (the reverse direction) every 30 seconds to keep centrifugal acceleration below the graviperception [73]. Dutch space agency updated a similar device with the help of the random walk principle as advised by Mesland, where the speed and direction of the motors changed at random time points [75]. Parameters of random rotation and direction are stored for the later analysis and repetition of the experiment [76].

A temperature-controlled room is required for experiments on mammalian cells, which are very sensitive to temperature fluctuation. Desktop RPM solves this problem. A miniaturized RPM with a maximum size of 50 cm × 50 cm × 50 cm allows the experiment to be done under a regular cell culture incubator [76]. Wuest et al. fitted the commercially available CO_2_ incubator onto the frames, in which temperature and other culture parameters are maintained and monitored through the incubator [77]. This device is called Random Position Incubator (RPI) [78]. Moreover, due to the suitable gas supply, this device also found application in 3-D tissue culture similar to RWV [77]. In RPI, frames are rotated at a constant speed, and the direction is changed at random time points [78]. In addition, a RPM called Microgravity Incubator (MGI) was built to run experiments on multiple samples simultaneously [42]. Similar to the desktop RPM, MGI can also be placed in a culture incubator [42].

The analysis during an experiment in space or simulated microgravity in real-time is as critical as cell culture condition. As described in the previous section, although microscopes are used on clinostats, most ground-based simulators are prone to vibration. Moreover, most experiments have been done with the fixation process, which cannot provide real-time observation. Pache et al. demonstrated Digital Holographic Microscopy (DHM) for real-time monitoring of cells in RPM [79]. DHM generates a hologram using interference, which is acquired by a digital camera [79]. Compared to conventional microscopy, DHM is non-invasive, label-free, and provides quantitate phase images [79]. Moreover, the same group combined DHM with widefield epifluorescence microscopy for more details of 3D cell morphology in simulated microgravity [80]. However, this method cannot be employed for desktop RPM [81]. Neelam et al. reported an image acquisition module comprises of a digital microscope with a magnification of 20× to 700×, a backlight to observe the sample, and a Wi-Fi module for streaming the acquired images in real time [81].

In clinostats, the sample holder size is kept in millimeter scale to minimize acceleration but limiting the capacity of the sample volume [82]. In RPMs, the addition of the vertical rotation axis generates acceleration in all directions, allowing for more sample volume [83]. Overall, RPM has more flexibility in selecting the parameters of the sample holder, which makes RPM more suitable for holding LOC devices. Przystupski et al. fabricated an all-glass LOC to investigate cancer cells in microgravity with a 3D clinostat [84]. In addition, Silvani et al. replicated the in vivo environment of glioblastoma multiforme (GBM) brain tumors with a microfluidic LOC device [85] and RPM for microgravity simulation [85]. Moreover, in a later work, Silvani utilized 3D printing for the fabrication of the LOC device called microgravity on chip (MOC) (Figure 5b) to cut costs and time for microgravity research for brain cancer cells [72]. This device also eliminates the conventional problems of bubble formation and leakage in RPM [72]. 

### 2.4. Diamagnetic Levitation

Diamagnetic levitation is another laboratory-based method to simulate microgravity on earth. While clinostats, RWV, and RPM cancel the gravity vector through rotation over the period, diamagnetic levitation counteracts the gravitational force (*F_g_*) by levitating the object with magnetic force (*F_m_*) [86]. The force *F*_*m*_ acting on an object is given as [86],
(1)$$\vec{F_{m}}=\frac{V\cdot \Delta c}{\mu_{0}}\left(\vec{B}\cdot \nabla \right)\vec{B}$$
where *V* is the volume of the object, Δ*c* is the magnetic susceptibility difference between the object and the surrounding medium, $\vec{B}$ is the magnetic flux density, $\mu_{0}$ is the magnetic permeability of free space, and $\left(\vec{B}\cdot \nabla \right)\vec{B}$ is the magnetic field gradient. Moreover, the acting gravitational force is given as [86],(2)$$\vec{F_{m}}=V\Delta \rho g$$where Δ*ρ* is the density difference between the object and surrounding medium and *g* is the gravitational acceleration. To levitate the object, magnetic force should equal gravitational force, which gives(3)$$\Delta \rho \cdot g=\frac{\Delta c}{\mu_{0}}\left(\vec{B}\cdot \nabla \right)\vec{B},$$

The above equation indicates that magnetic levitation of an object does not depend on the volume of the object. So large objects can be levitated through magnets. In addition, the field gradient generated by magnets and the magnetic susceptibility of the object play a significant role in magnetic levitation. Most biological organisms are diamagnetic and show homogeneity in their diamagnetic property, so levitation occurs at a molecular level and not only on the surface, allowing for possible simulation of microgravity [87]. However, magnetic force repels diamagnetic materials, when placed in the magnetic field gradient [88,89]. So, a strong magnetic field is required to counteract the gravity force and levitate the diamagnetic material [89].

A strong magnetic field can be produced with different types of electromagnets such as superconducting, bitter, or hybrid [90,91]. In a superconducting magnet, instead of using a ferromagnetic material coil to pass electric current, coil is cooled with liquid helium and has almost zero electric resistance. While bitter electromagnets have solenoids made of conducting disks to generate a high magnetic field. A hybrid magnet is a combination of both magnets which can produce a higher magnetic field than a bitter and superconducting magnet. However, Manzano et al. found genetic alteration of the *Arabidopsis*, and Glover et al. reported inhibition of Drosophila oogenesis in a strong magnetic field [92,93]. Moreover, Valiron et al. experimented to find the effect of a high magnetic field on different mammalian cells, such as fibroblasts, epithelial cells, and differentiating neurons [94]. Group found disorganization of cell assembly as well as cell loss in neurons [94].

Another approach to levitate object through magnetic force is keeping a diamagnetic object in a paramagnetic medium with high magnetic susceptibility and using a permanent magnet to levitate the object [95]. Tasoglu et al. developed a compact, label-free separation device, where a microcapillary tube containing suspended Red Blood Cells (RBCs) in a paramagnetic gadolinium-based (Gd^+^) medium was placed between two permanent magnets with the same pole opposing each other (Figure 7) [96]. Anil-Inevi et al. utilized the same device for cell culture and simulated longer-term microgravity conditions [95]. This approach is cost-effective, non-toxic, compact, and easy to set up, which is suitable for implementation on LOC devices to simulate microgravity in the lab [95]. Due to its advantages, Du et al. also utilized this approach to simulate microgravity in plant seeds [97]. The group developed a microfluidic chip to levitate *Arabidopsis* seeds and found a repressed auxin response in the absence of gravity [97].

## 3. CubeSat

Laboratory-based simulators successfully imitate microgravity conditions of space but cannot provide all parameters of the space environment (radiation, air composition, and launch stressors) [98]. CubeSat, a small autonomous cubic satellite, has attracted much attention due to its small size, low power consumption, and commercial off-the-shelf (COTS) components to carry out experiments in lower earth orbit (LEO) and deep space. The original purpose of building the CubeSat was to educate students and to attract awareness about space activities [99]. However, realizing its potential for applications in different fields, more than 1000 CubeSats have been launched since the development of the first satellite in 1999 [99]. California Polytechnic State University collaborated with Stanford University for the first CubeSat and set the standards for building the CubeSats [100]. The standard unit of CubeSat is defined as 1U which is a 10 cm cube (10 cm × 10 cm × 10 cm) with a mass of up to 1.33 kg [100]. From this standard, different form factors CubeSats, 2U, 3U, and 6U have been standardized [100]. Moreover, specifications for 12U and 27U are in the process of standardization for more extensive capabilities [100].

CubeSats can be classified into six categories: (i) earth science and space-borne application, (ii) deep space exploration, (iii) heliophysics: space weather, (iv) astrophysics, (v) space-borne in situ laboratories, (vi) technology demonstration, according to the primary objective of the mission [100]. The main objective of the space-borne in situ laboratory or lab-on-a-CubeSat is to carry out biological experiments in space and to control them from the ground. The heart of these types of CubeSats is the fluidic system to perform the necessary actions [101]. NASA’s Ames Research Centre is at the forefront of building CubeSats with biological experiments [101]. 

The first lab-on-a-CubeSat 3U GeneSat-1 was launched at the end of 2006, whose objective was to check the microgravity effect on two strains of *Escherichia coli* (*E*. *coli*) [102]. Subsequently, NASA developed two other 3U CubeSats PharmaSat and Organism/ Organic Exposure to Orbital Stresses (O/OREOS) in 2009 and 2010, respectively [103,104]. The PharmaSat’s mission was to measure the response of Saccharomyces cerevisiae (budding yeast) cells to an antifungal drug, while O/OREOS was the first CubeSat to carry out two experiments in one satellite [103,104]. The first experiment checked the microgravity effect on bacteria. The second experiment investigated the photodegradation of biomarkers and biological building blocks using UV-visible spectroscopy [104]. The 3U SporeSat, launched in 2014, was the first CubeSat with LOC devices to study the effect of microgravity on calcium signaling [105]. However, the experiment failed due to a problem with the illumination system [105]. All previously developed biological CubeSats had the 3U design. However, in 2017 EcAMSat was the first 6U (Figure 8a) CubeSat built to investigate the microgravity effect on dose-dependent antibiotic resistance of uropathogenic *E. coli* [106]. To date, all initiated biological CubeSats missions were in lower earth orbit. NASA is planning to launch the first deep space biological 6U CubeSat BioSentinel in August 2022 [107]. The main objective of BioSentinel is to explore the effect of ionizing radiation on Deoxyribonucleic acid (DNA) and cell damage response on growing yeast cells [108]. Integrated electrochemical detection of change in DNA and RNA can be achieved [109].

In contrast to the above SporeSat, other CubeSats have active fluidic control with the fluidic card. SporeSat has three LOC devices called biological discs (bioCDs), of which two are rotated to generate artificial gravity while one is kept stationary to expose to microgravity. GeneSat-1, PharmaSat, and EcAMSat have similarities in their fluidic system, but with updated designs. GeneSat-1 contains 12 wells (110 μL volume) in a fluidic card, while PharmaSat and EcAMSat expand fluidic card with 48 wells (100 μL volume) (Figure 8b). Furthermore, all devices have a flow path from the bottom to the top of the well and then to the waste bags, supported by external pumps and valves (Figure 8c). Instead of having one fluidic card system, O/OREOS has three different card systems with the same wells as GeneSat-1with 75 µL volume. O/OREOS does not have waste bags and has a hydrophobic filter on top to release air while filling the well. BioSentinel has the most advanced technology and has 18 fluidic cards with 16 100-μL wells in each card. In contrast to O/OREOS, BioSentinel is designed for fluid exchange to revive yeast and fluid flow from wells to waste lines and then to the designated waste bags.

CubeSat can operate in LEO and deep space without human presence, providing the highest quality of microgravity simulation. However, most CubeSats developments still handle fluid volume on the order of milliliters (mL) and have not yet implemented state-of-the-art LOC devices. As most LOC devices are built for ground base laboratories, there is a big potential for the deployment of this technology in CubeSats [110]. As example, recently Krakos et al. demonstrated all glass LOC device for fungi cultivation [111]. In which, group verified the use of LOC in microgravity using RWV and suggested use of this system for CubeSat mission [111].

## 4. Plausible Laboratory-Based Microgravity Simulators

Although research is advancing in the field of microgravity simulators as discussed above, novel laboratory-based simulators are needed to meet the increasing demand from space research. Mesland et al. developed Free Fall Machine (FFM) for long-term biological experiments and investigated cell cycle progression in Chlamydomonas, which showed similar results to that from space experiment [112]. The concept is to let the biological sample fall from a 1 m vertical guiding tube, providing 800 ms of microgravity and again send it back to the top, interrupting the microgravity for 50 ms. The interrupted period can be neglected as cells do not respond to altered gravity values for a short time. However, Schwarzenberg et al. failed to reproduce the same result of space experiments for human T-lymphocytes cells in FFM [113,114]. After this experiment, no other studies have been performed with FFM, so the suitability of the device for microgravity simulation is not validated. Further examination of this device is suggested by Ulbrich et al. in a review paper [115]. This concept can be implemented with more advanced technology and open more avenues for research.

The other possible device to simulate microgravity is the centrifuge. In the past, centrifuges are used to simulate hypergravity. Hypergravity is the term for gravity acceleration more than the value of the earth, e.g., 2 g, 5 g, 10 g, etc. However, Van Loon suggested a reduced gravity paradigm to simulate microgravity with the help of centrifuges [116]. The idea behind the reduced gravity paradigm (Figure 9a) is to let the biological system adapt to a hypergravity environment [116]. Once it adapts to this environment, reducing the gravity value to normal earth gravity and investigating the response generated by the two different acceleration levels [116]. The result could be the same as the microgravity response [117]. However, to implement this paradigm, the cell culture system needs to be sturdy at the hypergravity level. Moreover, some studies have been done with this paradigm [118,119], but a systematic approach is required to validate the concept.

Earth gravity can be countered using external force, such as the magnetic force discussed in the diamagnetic levitation method. However, levitation can also be achieved with other forces, including electrostatic, aerodynamic, optical, and acoustic (ultrasonic sound). These levitation techniques have been utilized in container-free processes on earth and space. However, studies to simulate microgravity with these techniques are restricted or not available. Chang and Trinh successfully grew lysozyme and thaumatin crystals by generating a lower-gravity environment on earth using combined electrostatic (Figure 9b) and ultrasonic (Figure 9c) levitation methods [120]. In this device, surface charged bearing protein solution droplet is levitated by electrostatic force. The droplet is rotated around the horizontal axis under ultrasonic streaming and radiation pressure to reduce the gravity effect [120].
Figure 9Principle of different possible microgravity simulators. (**a**) Reduce gravity paradigm. (Adapted with permission from [116]. Copyright 2016, Van Loon) (**b**) Electrostatic levitation. (**c**) Acoustic levitation. (**d**) Aerodynamic levitation. (Adapted with permission from [121]. Copyright 2021, Authors).
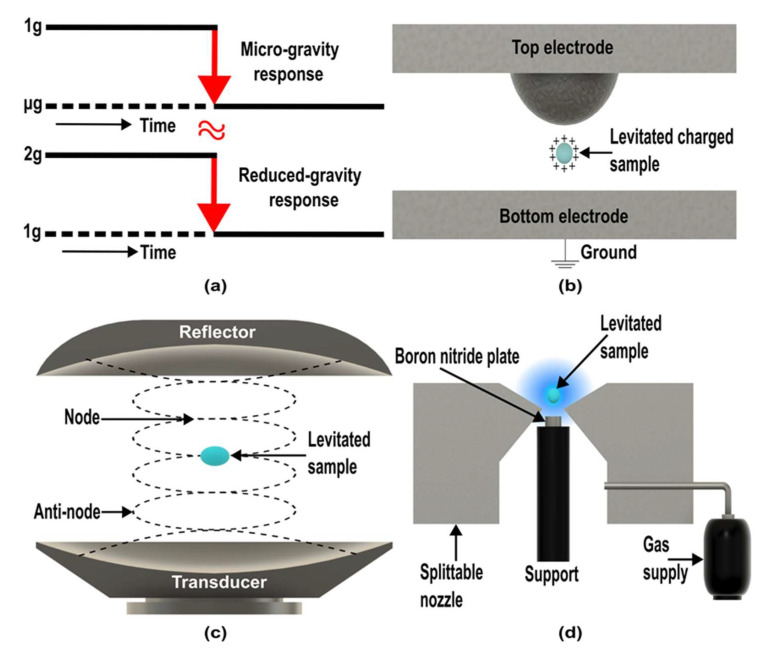


Cao et al. obtained crystals of NaCl, NH_4_Cl, lysozyme, and proteinase K rapidly with acoustic levitation [122]. The result exhibited better growth compared to conventional methods [122]. Moreover, the group considered acoustic levitation a valuable ground-based microgravity simulator, which can perform crystallization and screen the crystallization condition in space [122]. However, no further research has been found in this direction to simulate microgravity with acoustic levitation. Li et al. analyzed early zebrafish embryos with acoustic levitation and found that the embryo’s otolith cannot feel environment sound [123]. Moreover, being suspended in a water droplet, otolith does not perceive the earth’s gravity [123]. These effects lead to slow and abnormal growth of the embryo’s otolith [123]. However, differentiation of both effect and comparison with real space environment is required to consider acoustic levitation as a valid ground-based microgravity simulator.

Moreover, Sun et al. developed a bounce-drop method to determine surface tension with the help of aerodynamic levitation (ADL) (Figure 9d) [121]. This technique allowed stably levitated droplets to fall on a boron nitride surface below the splitable nozzle [121]. While bouncing back, the droplet oscillation is excited without any external force and experiences free-fall conditions [121]. The surface tension data of liquid gold matched the data taken in microgravity [121]. Later, the same group used Front Tracking (FT) simulation technique with the drop bounce technique to simulate and measure surface tension in molten Al_2_O_3_ and validated the technique to simulate microgravity conditions with ADL on the ground [124]. However, these levitation techniques are not widely available to the research community, which restricts the exploration of these devices to simulate microgravity.

## 5. Summary and Conclusions

The first section of the review describes the conventional laboratory-based microgravity simulators and their advancements with the implementation of LOC technology. Detailed information about laboratory-based devices can be found in Table 1. Apart from microgravity simulation, these devices can be used for hypergravity and partial gravity [125]. The term of partial gravity describes the gravity magnitude between zero to earth gravity level, e.g., Moon (0.17 g), Mars (0.38 g). Researchers inclined a clinostat [126] and constructed a centrifuge clinostat [127] to generate partial gravity. Moreover, Manzano et al. developed two novel partial gravity paradigms with the help of RPM. First, the group modified RPM hardware by including a centrifuge. Next, software protocols were changed to control the motor RPM [127]. Apart from clinostat and RPM, the gravity magnitude can also be altered by diamagnetic levitation by changing the value of the magnetic field gradient [128].

RPM and clinostat simulate microgravity by changing the direction of the gravity vector, so the result must be interpreted carefully. In diamagnetic levitation, the strong magnetic field may affect the biological sample. So, careful design is required to distinguish the microgravity effect from the magnetic field effect. In addition, the achieved quality of microgravity on earth is lower than the microgravity value in LEO and deep space (Table 1). The second section of the review explores the small cubic satellite. CubeSat is not a laboratory-based device but can be controlled from an earth-based laboratory. Moreover, the reason to include CubeSat into this review is the utilization of LOC technologies to carry out experiments in LEO and deep space. Moreover, the results achieved with CubeSat are more accurate than the laboratory-based simulators. 

In space, an experiment cannot be initialized until the deployment phase. The sample stays at room temperature, restricting the experiment on mammalian cells or sensitive biological samples as they require elevated temperatures for cell culture [129]. In addition, the overall duration to receive the results of the experiments is relatively high compared to laboratory-based devices. Due to these reasons, although more than a thousand CubeSats have been launched, only six CubeSats are designed for life science experiments and developed by a single research group at NASA. 

Due to the increasing interest in space research, new devices and methods are required to simulate microgravity in a laboratory and meet growing demand. The third section of the review follows through different possible methods to simulate microgravity in the lab. However, research with these techniques is significantly underreported. Results obtained with these devices do not match and thus limit their validation as microgravity simulators on earth. Although only few information is available on these devices, it provides an excellent opportunity to think out of the box, leading to a better way to simulate microgravity on earth. 

Conventional devices for simulating microgravity are large, expensive, and only available to research groups connected with space agencies. This drawback can be overcome by utilizing miniaturization such as LOC technology in a microgravity simulator. However, compared to the advancement of LOC devices in other fields, less attention has been paid to microgravity or its simulation on earth. Moreover, no previous reviews have been done on this topic as well. The present review fills this gap and provides a comprehensive overview of different methods leading to the use of LOC devices to perform microgravity experiments in the laboratory. In addition, this review also explores novel devices to simulate microgravity that has not been reviewed previously. 

## Figures and Tables

**Figure 1 micromachines-14-00116-f001:**
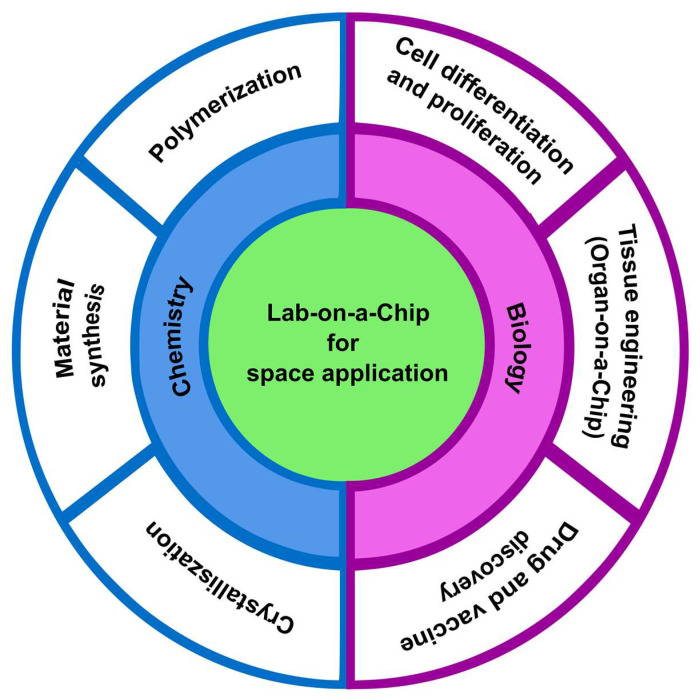
Lab-on-a-chip for biological and chemical space applications.

**Figure 2 micromachines-14-00116-f002:**
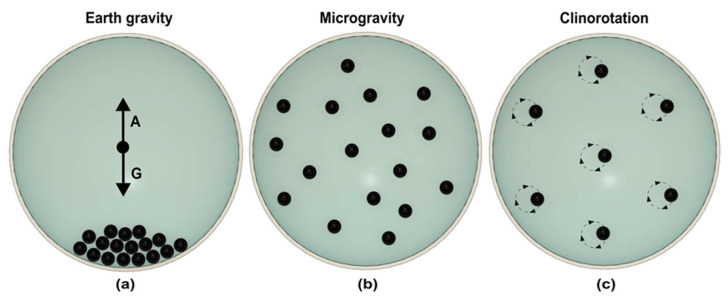
Schematic representation of fast rotating clinostat. (**a**) On earth, particles sediment in the sample holder. The downward gravity (G) and buoyancy (A) forces determine the particle’s position. (**b**) In microgravity, particles are distributed homogeneously due to the lack of gravitational force. (**c**) Fast rotation of sample holder perpendicular to gravity vector generates circular motion of particles. At appropriate speed, no relative circular motion of the particle is visible, generating the same situation as microgravity. (Adapted with permission from [29]. Copyright 2005, Cambridge University Press).

**Figure 3 micromachines-14-00116-f003:**
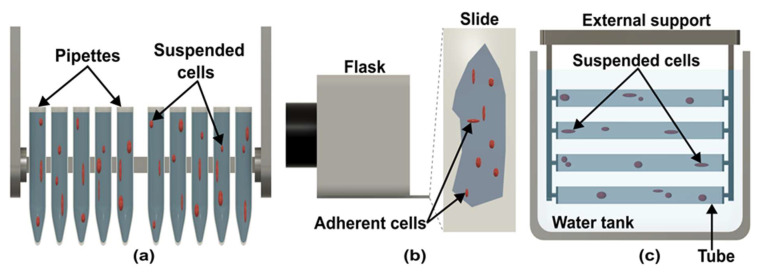
Different designs of clinostat. (**a**) Pipette/cuvette clinostat used for suspended cells. (**b**) Slide-flask used for adherent cells in slide-flask clinostat. (**c**) Scheme of submerged clinostat.

**Figure 4 micromachines-14-00116-f004:**
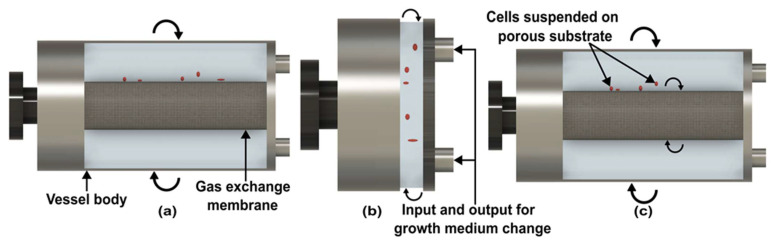
Side-view of rotating wall vessels. (**a**) Slow turning lateral vessel. (**b**) High aspect ratio vessel. (**c**) Rotating wall perfused vessel. (Adapted with permission from [58]. Copyright 2005, Elsevier Ltd.).

**Figure 5 micromachines-14-00116-f005:**
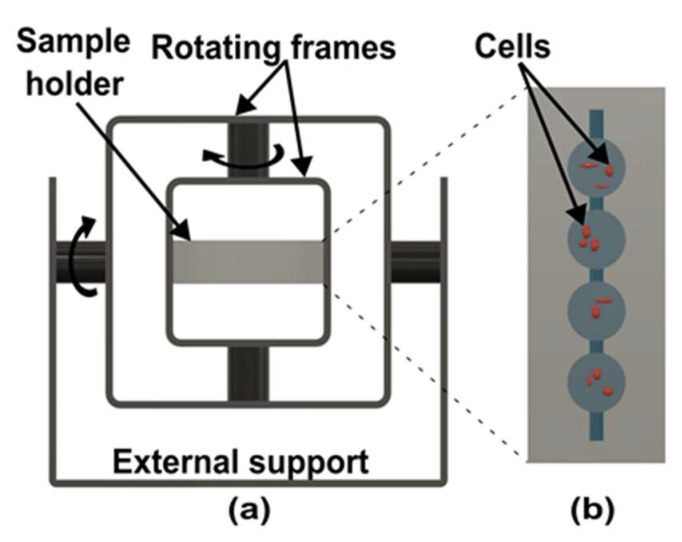
Schematic of random position machine. (**a**) Design of random position machine with two rotational axes. (Adapted with permission from [20]. Copyright 2020, Authors) (**b**) Top view of microgravity on-chip to research brain cancer cells. (Adapted with permission from [72], Copyright 2022, Authors).

**Figure 6 micromachines-14-00116-f006:**
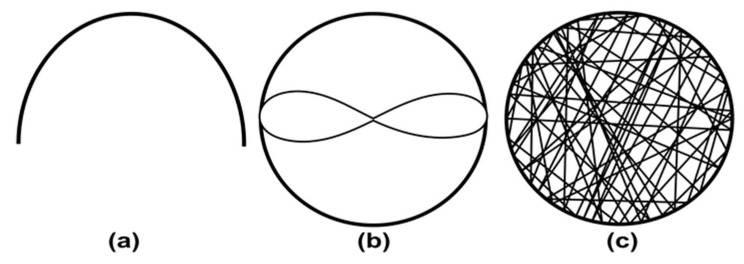
Sample motion in random position machine. (**a**) Both motors at the same speed. (**b**) motor speed rate 1:2. (**c**) Both motors are at random speeds. (Adapted with permission from [73]. Copyright 1992, The Botanical Society of Japan).

**Figure 7 micromachines-14-00116-f007:**
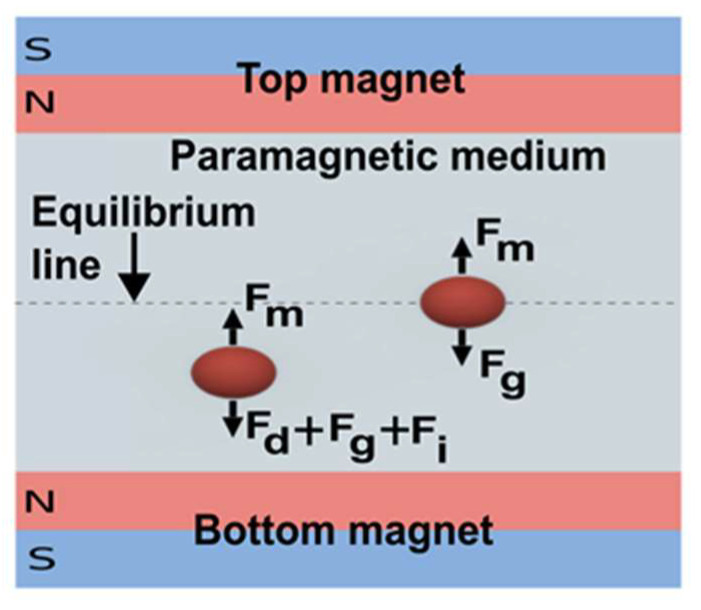
Scheme of diamagnetic levitation with a permanent magnet. (*F_m_* = magnetic force, *F_g_* = Gravitational force, *F_d_* = drag force, *F_i_* = inertial force) (Adapted with permission from [96]. Copyright 2015, WILEY-VCH Verlag GmbH & Co. KGaA, Weinheim).

**Figure 8 micromachines-14-00116-f008:**
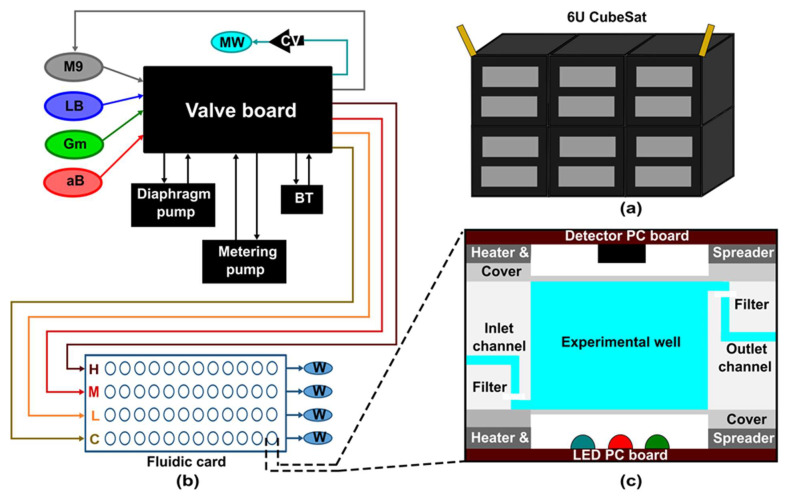
EcAMSat CubeSat design and fluidic system. (**a**) Schematic design presentation of 6U CubeSat. (**b**) The simplified fluidic system of EcAMSat. BT = bubble trap; CV = check valve; MW = main waste bag; W = card waste bag; C, L, M, H = control, low, medium, and high banks of the card, referring to the relative does of Gm delivered to each bank. (**c**) Cross-section of the experimental well in the fluidic system of EcAMSat. (Adapted with permission from [106]. Copyright 2020, Elsevier Ltd.).

**Table 1 micromachines-14-00116-t001:** Detailed information about laboratory-based microgravity simulators.

	Type	Features	Simulation Technique	Microgravity Quality	Microgravity Duration
**Clinostat**	Cuvette/Pipette	Shape—Cylinder	Rotation	≤10^−3^ g	Hours to Weeks
		Diameter—3.5 mm			
	Slide Flask	Shape—Rectangle			
		Width—9 cm			
		Length—9 cm			
	Submersed	Shape—Cylinder			
		Diameter—4.1 mm			
	PMT	Shape—Cylinder			
		Diameter—4 mm			
		Length—5 cm			
	Microscope	Shape—Cylinder			
		Diameter—30 mm			
**RWV**	STLV	Shape—Cylinder	Rotation	≤10^−3^ g	Hours to Weeks
		Diameter—9.5 cm			
		Length—9.6 cm			
	HARV	Shape—Cylinder			
		Diameter—12.7 cm			
		Length—0.64 cm			
	RWPV	Shape—Cylinder			
		Diameter—5 cm			
		Length—7 cm			
**RPM**	Desktop RPM	Shape—Cubic	Rotation	10^−4^ g	Hours to Weeks
	RPI				
	MGI				
**Diamagnetic Levitation**	Bitter Magnet	Features can vary according to the experiment	Magnetic Force	<10^−2^ g	Minutes to Hours
	Superconducting Magnet				
	Permanent Magnet				
**CubeSat**	GeneSat-1	CubeSat Size—3U		10^−6^ g	21 days
		Payload Size—2U			
		Weight—6.8 kg			
	PharmaSat	CubeSat Size—3U			>21 days
		Payload Size—2U			
		Weight—5.5 kg			
	O/OREOS	CubeSat Size—3U			6 months
		Payload Size—1U			
		Weight—5.5 kg			
	SporeSat	CubeSat Size—3U			Not Given
		Payload Size—2U			
		Weight—5.5kg			
	EcAMSat	CubeSat Size—6U			>120 days
		Payload Size—3U			
		Weight—14 kg			
	BioSentinel	CubeSat Size—6U			6–12 months
		Payload Size—4U			
		Weight—14 kg			

## Data Availability

Not applicable.

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
