# Peer review of "Lab-on-a-Chip Technologies for Microgravity Simulation and Space Applications"

_micromachines, 2022, doi:10.3390/mi14010116_

Round 1
Reviewer 1 Report
The review article is well written by properly covering the major technical aspects of the LOC and their potential use for space applications. I am having minor comments before considering it for possible application.
1) Please include a figure which briefly provides various space applications in LOC.
2) Figure 2 subsection numbering (a,b,c) should be either upside or downside. Please make them regulated. Also, amend the same for other figures.
3) Figure 3 is repeating!
4) Please renumber the figures in citations as well as captions
Author Response
1) We added the figure which include important space applications, which can benefit from LOC 2) Changed as suggested 3) Originally figure 3 now figure 4 depicts the different type of Rotating Wall Vessels. Figure 4a is STLV and 4c is RWPV. There is no design difference between them. But STLV has only vessel body rotation, while RWPV has vessel body as well as gas exchange membrane rotation. So, we put the rotation sign at both part in RWPV while in STLV has only one rotation sign. 4) Changed as suggestedReviewer 2 Report
The manuscript entitled "Lab-On-a-Chip Technologies for Microgravity Simulation and Space Applications" is a well-written knowledge compendium referring to the LOC-based instruments and their utility for space applications. The manuscript covers both the research in simulated microgravity environments utilizing microgravity simulators, as well as in real space conditions with CubeSat nanosatellites. The Authors have also mentioned new trends towards microgravity imitation by means of aerodynamic levitation or hypergravity reduction to normal gravity. The paper is highly interesting and I truly liked it. Only minor points have been noticed and are presented as follows:
1. In the Abstract the statement “However, the search for life outside the earth requires more intensive research.” is more about the astrobiology, rather than space biology, for which LOCs are majorly used.
2. In the Introduction part, please also mention about the blow in 3D printing technique that is recently used for LOC fabrication. It significantly cut the costs, time and does not require well-equipped cleanroom laboratories.
3. Fig. 1 could be placed below the mentioning text.
4. In a single paragraph sometimes both tenses – Present Simple and Past Simple are used – please check.
5. Caption of the Fig. 4 is on the other page than the Figure.
6. Maybe you could use the same nomenclature for magnetic force Fmag/Fm .
7. Line 396 – Genesis-1 or GeneSat-1? In this paragraph there is also a problem with aligning.
8. You can also add to your consideration the research about studies of fungi utilizing microgravity simulator and LOC platform – “Krakos, A.; Śniadek, P.; Jurga, M.; Białas, M.; Kaczmarek-Pieńczewska, A.; Matkowski, K.; Walczak, R.; Dziuban, J. Lab-on-Chip Culturing System for Fungi—Towards Nanosatellite Missions. Appl. Sci. 2022, 12, 10627. https://doi.org/10.3390/app122010627”
9. In the last paragraph, there is a statement that no previous reviews have been done about the LOCs for space research, which actually is partially truth. There have been recently published some works on this topic, e.g.:
https://www.sciencedirect.com/science/article/pii/S0956566322008600?via%3Dihub
https://www.sciencedirect.com/science/article/pii/S2214552422000682?via%3Dihub
https://www.frontiersin.org/articles/10.3389/frspt.2021.779696/full
Author Response
1) Changed the sentence to make it more space biology based instead of astrobiology.
2) Include the advantage of 3D- printing technologies from line 55-58.
3) Changed as suggested.
4) Can see the changes through Trackchange
5) Changed as suggested.
6) Changed it to Fm
7) Changed it to GeneSat and aligned the paragraph.
8) Added the reference in CubeSat section line 411-414.
9) Most of the past reviews are to develop LOC devices (mostly for tissue engineering) for space application, e.g. in International Space Station. However, there is no review that describe the use of LOC device to simulate microgravity on earth. That is why we selected it as our topic.